# Effect of High-Dose Intravenous Vitamin C on Postpartum Oxidative Stress in Severe Preeclampsia

**Monika Korenc** [1], **Joško Osredkar** [2], **Ksenija Gersak** [3], **Kristina Kumer** [2], **Teja Fabjan** [2], **Sasa Sterpin** [2] **and Miha Lucovnik** [1,3,*]

[1]  Department of Perinatology, Division of Obstetrics and Gynecology, University Medical Center Ljubljana, Slajmerjeva 4, 1000 Ljubljana, Slovenia; mony.korencek@gmail.com

[2]  Institute of Clinical Chemistry and Biochemistry, University Medical Center Ljubljana, Njegoseva 4, 1000 Ljubljana, Slovenia; josko.osredkar@kclj.si (J.O.); kristina.kumer@kclj.si (K.K.); teja.fabjan@kclj.si (T.F.); sasa.sterpin@kclj.si (S.S.)

[3]  Faculty of Medicine, University of Ljubljana, Vrazov trg 2, 1000 Ljubljana, Slovenia; ksenija.gersak@mf.uni-lj.si

*  Correspondence: mihalucovnik@yahoo.com; Tel.: +386-31-318-681

**Abstract:** Purpose: To determine whether high-dose intravenous vitamin C reduces oxidative stress in patients with severe preeclampsia in the first days postpartum. Methods: Biomarkers of oxidative stress were assessed as secondary outcomes of a single-center, randomized, placebo-controlled trial. Thirty-four patients with singleton pregnancies complicated by severe features of preeclampsia were randomized into two groups: intravenous vitamin C (1.5 g/6 h) ($n = 17$) or placebo ($n = 17$). Urinary concentrations of dityrosine, 8-hydroxy-2-deoxyguanosine (8-OHdg), 8-isoprostane, and N epsilon-(hexanoyl) lysine (HEL) were measured at days one and three after delivery and normalized for urinary creatinine in 22 of patients included (12 in vitamin C and 10 in placebo group). The Mann–Whitney U-test was used to compare values of oxidative stress biomarkers at days one and three after delivery in vitamin C vs. placebo groups ($p \leq 0.05$ significant). Results: Dityrosine and 8-OHdg values did not differ significantly between the two study groups at day one after delivery ($p = 0.23$ and $p = 0.77$, respectively), but were significantly lower in the vitamin C group compared to the placebo group at day three after delivery ($p = 0.04$ and $p = 0.03$, respectively). Values of 8-isoprostane and HEL did not differ significantly between the two study groups at day one ($p = 0.41$ and $p = 0.42$, respectively), as well as at day three, after delivery ($p = 0.25$ and $p = 0.24$, respectively). Conclusion: High-dose intravenous vitamin C treatments in patients with severe preeclampsia reduced urinary levels of dityrosine and 8-OHdg (markers of protein and DNA oxidative damage, respectively) on day three after delivery. Vitamin C treatment had no significant effect on lipid peroxidation biomarkers, i.e., 8-isoprostane and HEL.

**Keywords:** ascorbic acid; oxidative stress; preeclampsia; vitamin C

## 1. Introduction

Preeclampsia, a pregnancy-specific multisystem disorder characterized by new onset hypertension and proteinuria or end-organ dysfunction after 20 weeks of gestation, affects 2% to 5% of pregnancies [1–3]. It is associated with a high risk of fetal and neonatal complications, as well as maternal morbidity and mortality [3–5]. The exact pathophysiology of preeclampsia remains incompletely understood, but increased oxidative stress, i.e., an imbalance between the formation of reactive oxygen species (ROS) and antioxidant defense mechanisms, may contribute significantly to disease development and its clinical course [6,7]. Placental malperfusion leads to the enhanced

release of ROS and lower levels of antioxidants have been found in preeclamptic patients [8–10]. Targeting excessive oxidative stress with antioxidant therapy could potentially reduce the risk of preeclampsia-related adverse outcomes.

Vitamin C is the major water-soluble antioxidant present within the cells and extracellular fluids [11]. High-dose intravenous treatment with vitamin C has been shown to improve outcomes in patients with acute respiratory distress syndrome, severe burns, septic shock, and surgical intensive care unit patients [12–20]. One of the proposed mechanisms behind the beneficial effects of vitamin C in critically ill patients is a reduction in oxidative stress [21]. Oxidative stress cannot be measured by the direct detection of ROS due to their very short half-life [22]. Instead, the amount of oxidative stress can be assessed by measuring stable products of non-enzymatic reactions between biological molecules (DNA, proteins, and lipids) and ROS [23,24]. Urinary concentrations of dityrosine, 8-hydroxy-2-deoxyguanosine (8-OHdg), 8-isoprostane, and N epsilon-(hexanoyl) lysine (HEL) have been used for this purpose and are well-validated biomarkers for the evaluation of oxidative stress [24,25]. No studies concerning the effects of vitamin C treatment on oxidative stress in postpartum patients with preeclampsia have been published so far.

Most studies on prophylactic supplementation with oral antioxidants, including 1 g vitamin C daily, showed no significant reduction in the risks of preeclampsia, intrauterine growth restriction, or the risk of death or other serious outcomes in mothers and infants [16–18]. However, there are no studies to date on the use of high intravenous doses of vitamin C for tertiary prevention in preeclampsia patients, i.e., the prevention of serious complications once the disease has already occurred. Given the safety and low cost of vitamin C, this could be a promising approach to reducing the risks of severe maternal morbidity in pregnant and postpartum patients with severe forms of preeclampsia.

This study aimed to determine whether high-dose intravenous vitamin C administration reduces oxidative stress in patients with severe preeclampsia in the first days postpartum.

## 2. Methods

Biomarkers of oxidative stress were assessed as secondary outcomes of a single-center, randomized, placebo-controlled, double-blind trial. The trial was conducted at the Department of Perinatology of the University Medical Centre Ljubljana, Slovenia, from April 2018 to June 2019. All women included in the study provided written informed consent for study participation. The National Medical Ethics Committee approved the study (Project number 0120-527/2017/4, approved on 12 September 2017). The trial is registered at ClinicalTrials.gov with the identifier NCT03451266.

Thirty-four consecutively admitted patients with singleton pregnancies complicated by severe preeclampsia were included in the study at hospital admission. Severe preeclampsia was defined by severe features of the disease using the American College of Obstetricians and Gynecologist Task Force on Hypertension in Pregnancy recommendations: new onset cerebral or visual disturbances; pulmonary edema; thrombocytopenia (platelet count <100,000/μL); elevated liver enzymes (transaminases) to twice the normal upper limit; severe persistent pain in the right upper or middle upper abdomen that does not respond to medication and is not explained by another condition or both; renal insufficiency (serum creatinine >97 μmol/L), or the doubling of serum creatinine in the absence of other renal diseases; systolic blood pressure ≥160 mm Hg or a diastolic blood pressure ≥110 mm Hg measured on more than one occasion at least 4 h apart while the patient is on bed rest (unless antihypertensive therapy was initiated before this time) [4].

After written informed consent, patients were randomized in a 1:1 ratio into two groups. They received either 1.5 g of intravenous vitamin C in 100 mL 0.9% NaCl within 30 min of delivery and then every 6 h for the first 72 h postpartum or a placebo (100 mL of intravenous 0.9% NaCl within 30 min of delivery and then every 6 h for the first 72 h postpartum).

Randomization was done with sequentially numbered, opaque, computer-generated sealed envelopes. Randomization codes were kept at the pharmacy department where the treatment, (vitamin C and 0.9% NaCl) was prepared. High-dependency unit (HDU) physicians (anesthesiologists

and perinatologists), patients, and study investigators were blinded to the type of medication administered by nurses, who prepared and administered the study medications.

Although the safety of vitamin C has been well established even at high doses, a theoretical concern regarding vitamin C is that it may be metabolized into oxalic acid, leading to calcium oxalate nephropathy [13]. Therefore, information on potential side effects of vitamin C administration, especially related to the occurrence of calcium oxalate nephropathy, was collected and analyzed. We also collected information on any other complications that might be theoretically associated with vitamin C treatment (as an "open question" in our data collection protocol).

## 2.1. Oxidative Stress Biomarker Measurements

Urine was collected at day one after delivery (within 24 h from delivery) and at day three after delivery (48 to 72 h from delivery) in 22 of 34 patients included in the study (12 in vitamin C and 10 in placebo group) (Figure 1).

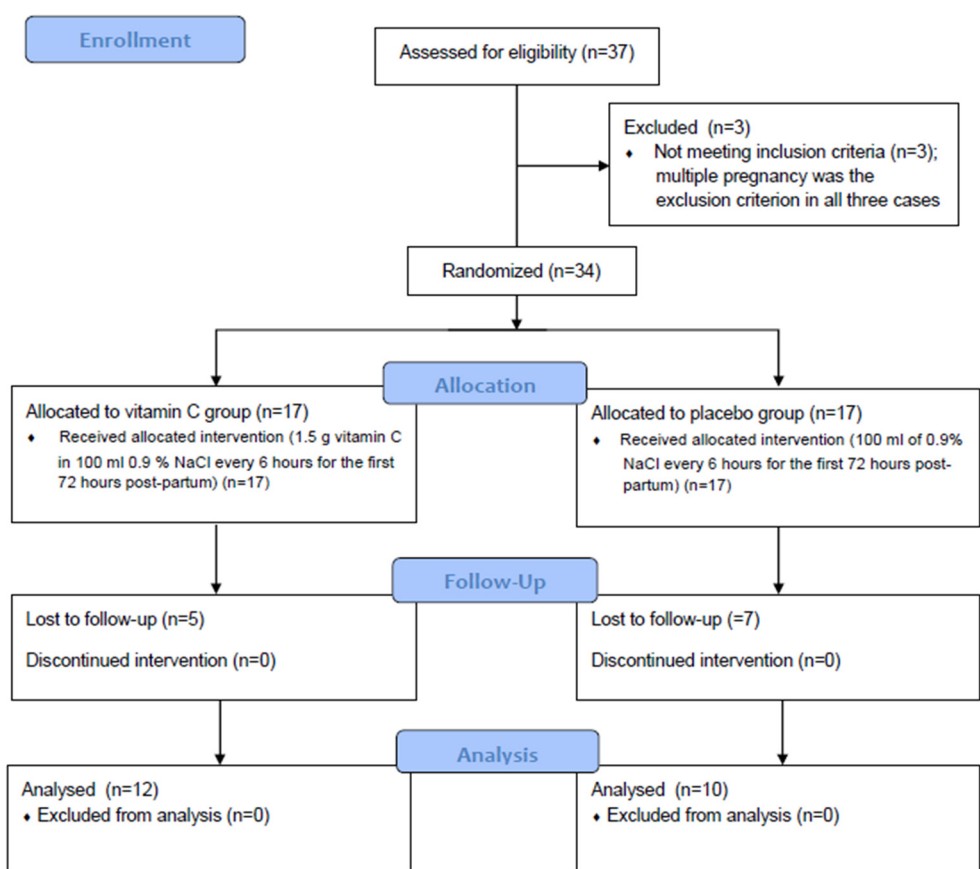

**Figure 1.** Randomization and follow up of study participants.

Spot mid-stream urine samples were collected in sterile cups and stored in the laboratory at −80 °C until they were assayed. Previous studies have shown stable levels of oxidative stress biomarkers at −80 °C [24,26,27]. For the determination of three of the four biochemical parameters, we used commercial ELISA kits: 8-OHdG Check ELISA, HEL ELISA and the DT ELISA kit provided by the Japanese Institute for the Control of Aging (JaICA, Tokyo, Japan). We determined 8-izoprostane using a kit from IBL International (Hamburg, Germany). Concentrations of all four biomarkers, i.e., dityrosine, 8-OHdg, 8-isoprostane, and HEL, were normalized for urinary creatinine. Urinary creatinine was obtained using the Roche reagent kit on a Roche Modular P analyzer (Roche Diagnostics GmbH, Mannheim, Germany). All tests were performed following the instructions of the kit provider.

## 2.2. Statistical Analysis

For continuous variables, data were expressed as the median together with the 25th and 75th percentiles (interquartile range). Categorical data were summarized as frequencies and percentages. For comparison between the two study groups (placebo vs. vitamin C), a Mann–Whitney U-test was used for continuous variables and a chi-square test for categorical variables. Values of oxidative stress biomarkers at day one after delivery vs. day three after delivery within each group were compared using a Mann–Whitney U-test. For all tests, a two-tailed *p*-value ≤ 0.05 was considered statistically significant. The software used for statistical analysis was IBM SPSS Statistics for Windows, Version 25.0 (Armonk, NY, USA: IBM Corp.).

## 3. Results

Thirty-seven patients admitted to the perinatal high-dependency unit due to a diagnosis of severe preeclampsia were assessed for eligibility during the study period. Three were pregnant with twins and, therefore, not included. Thirty-four fulfilled the inclusion criteria and all consented to participate in the study. They were randomly allocated to vitamin C or placebo groups. Urine samples were not collected due to organizational/logistic issues in two patients (one in the vitamin C group and one in the placebo group). For the same reasons, measurements of oxidative stress biomarkers and/or urinary creatinine were not performed in ten patients (four in the vitamin C group and six in the placebo group) (Figure 1). All urine samples on day one were collected within the first six hours after delivery, i.e., after patients received a single dose of study medication (vitamin C or placebo).

The baseline characteristics of women and their neonates included in the two study groups were similar (Table 1). Table 1 also presents severe features of preeclampsia meeting the inclusion criteria that were also evenly distributed between the groups.

**Table 1.** Baseline characteristics of study participants.

| Characteristic | Vitamin C (*n* = 12) | Placebo (*n* = 10) | *p*-Value |
|---|---|---|---|
| Maternal age (years) | 30 (22–41) | 30 (23–40) | 0.97 |
| Pre-pregnancy BMI (kg/m$^2$) | 23 (17–35) | 26 (18–41) | 0.23 |
| BMI at delivery (kg/m$^2$) | 29 (22–56) | 31 (20–49) | 0.69 |
| Nulliparity | 10 (83%) | 7 (70%) | 0.28 |
| Gestational age (weeks) | 33 (24–38) | 33 (26–39) | 0.38 |
| Caesarean delivery | 9 (75%) | 9 (90%) | 0.69 |
| Neonatal birth weight (g) | 1465 (490–3110) | 1820 (630–3870) | 0.33 |
| SGA | 8 (67%) | 4 (40%) | 0.15 |
| Systolic blood pressure ≥160 mm Hg at inclusion | 12 (100%) | 10 (100%) | / |
| Diastolic blood pressure ≥110 mm Hg at inclusion | 12 (100%) | 10 (100%) | / |
| Neurological symptoms | 3 (25%) | 4 (40%) | 0.21 |
| Serum creatinine >97 μmol/L | 1 (8%) | 0 (0%) | 0.33 |
| Elevated liver enzymes and/or pain in the right upper or middle upper abdomen | 8 (67%) | 3 (30%) | 0.09 |
| Thrombocytopenia | 3 (25%) | 2 (20%) | 0.78 |
| Urine output at inclusion (mL/kg/h) | 0.62 (0.40–1.17) | 0.52 (0.27–1.33) | 0.42 |
| Serum creatinine at inclusion (μmol/L) | 59 (48–133) | 55 (43–67) | 0.35 |
| sFlt-1/PlGF | 268 (152–2324) | 399 (111–998) | 0.12 |

Data are presented as median (range) or *n* (%). *p*-value calculated by Mann-Whitney U-test or chi-square test; body mass index (BMI); small for gestational age (SGA; = birthweight < 5th centile for gestational age using population-specific growth curves); liver enzymes levels of aspartate transaminase (AST) and/or alanine transaminase (ALT) elevated to twice the normal upper limit; thrombocytopenia platelet count <100,000/μL; soluble fms-like tyrosine kinase-1 (sFlt-1); placental growth factor (PlGF).

Dityrosine values did not differ significantly between the two study groups at day one after delivery (*p* = 0.23), but were significantly lower in the vitamin C group compared to the placebo group at day three after delivery (*p* = 0.04) (Figure 2). Differences in dityrosine values between day one and day three after delivery reached statistical significance in neither the placebo group (*p* = 0.46) nor the vitamin C group (*p* = 0.22).

Values of 8-OHdg did not differ significantly between the two study groups at day one after delivery (*p* = 0.77), but were significantly lower in the vitamin C group compared to the placebo group at day three after delivery (*p* = 0.03) (Figure 3). Differences in 8-OHdg values between day one and day three after delivery reached statistical significance in neither the placebo group (*p* = 0.26) nor the vitamin C group (*p* = 0.26).

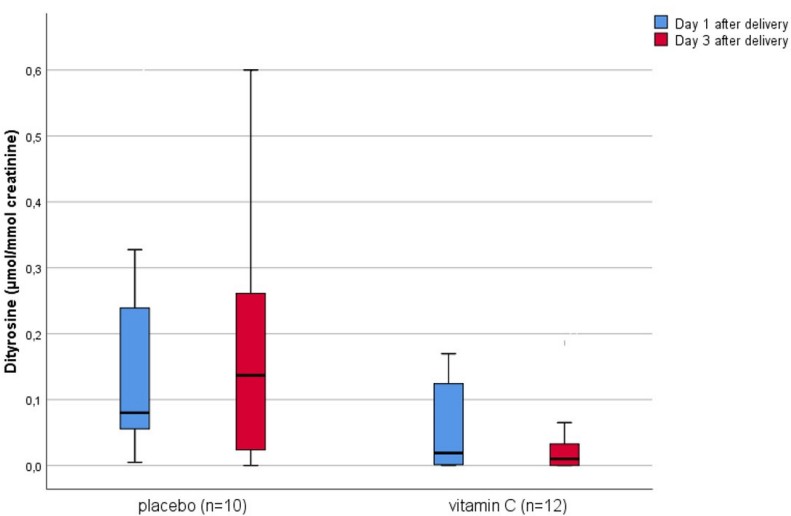

**Figure 2.** Comparison of dityrosine values in vitamin C and placebo groups. Dityrosine values did not differ significantly between the two study groups at day one after delivery (*p* = 0.23), but were significantly lower in the vitamin C group compared to the placebo group at day three after delivery (*p* = 0.04). Data are presented in box-and-whisker plots with the median and interquartile range and minimum and maximum data points.

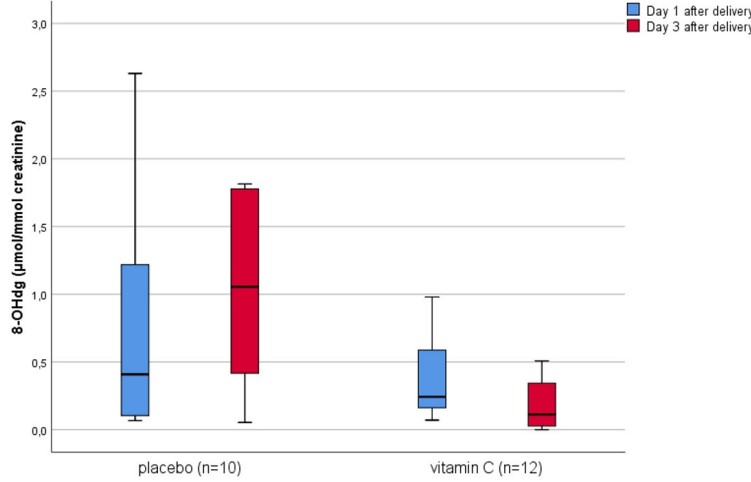

**Figure 3.** Comparison of 8-hydroxy-2-deoxyguanosine (8-OHdg) values in vitamin C and placebo groups. Values of 8-OHdg did not differ significantly between the two study groups at day one after delivery (*p* = 0.77), but were significantly lower in the vitamin C group compared to the placebo group at day three after delivery (*p* = 0.03). Data are presented in box-and-whisker plots with the median and interquartile range and minimum and maximum data points.

Values of 8-isoprostane did not differ significantly between the two study groups at day one as well as at day three after delivery ($p = 0.41$ and $p = 0.25$, respectively) (Figure 4). Differences in 8-isoprostane values between day one and day three after delivery reached statistical significance in neither the placebo group ($p = 0.26$) nor the vitamin C group ($p = 0.39$).

HEL values did not differ significantly between the two study groups at day one as well as at day three after delivery ($p = 0.42$ and $p = 0.24$, respectively) (Figure 5). Differences in HEL values between day one and day three after delivery did not reach statistical significance in the placebo group ($p = 0.19$), but were significantly higher at day one compared to day three in the vitamin C group ($p = 0.05$).

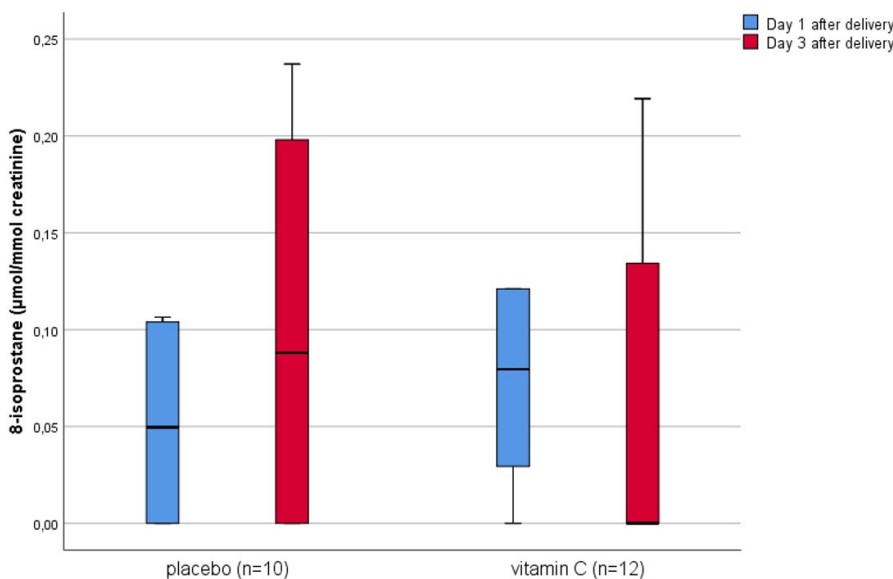

**Figure 4.** Comparison of 8-isoprostane values in vitamin C and placebo groups. Values of 8-isoprostane did not differ significantly between the two study groups at day one as well as at day three after delivery ($p = 0.41$ and $p = 0.25$, respectively). Data are presented in box-and-whisker plots with the median and interquartile range and minimum and maximum data points.

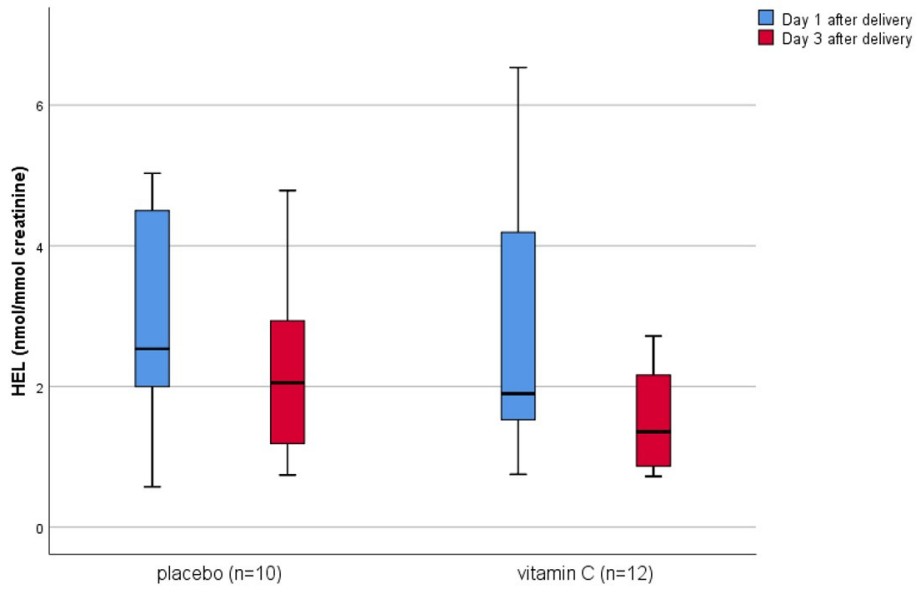

**Figure 5.** Comparison of N epsilon-(hexanoyl) lysine (HEL) values in vitamin C and placebo groups. HEL values did not differ significantly between the two study groups at day one as well as at day three after delivery ($p = 0.42$ and $p = 0.24$, respectively). Data are presented in box-and-whisker plots with the median and interquartile range and minimum and maximum data points.

None of the patients in the study developed oxalate renal calculi, which have been reported as a potential adverse effect of high-dose vitamin C treatment in burn patients [13]. We also found no other adverse effects that could be associated with vitamin C treatment.

## 4. Discussion

The main finding of this study is that high-dose intravenous vitamin C treatment started postpartum in patients with severe preeclampsia reduced urinary levels of dityrosine and 8-OHdg (markers of protein and DNA oxidative damage, respectively) on day three after delivery. On the first day after delivery, none of the oxidative stress biomarkers studied differed significantly between the placebo and vitamin C groups. It has to be noted, however, that all measurements on day one were done within six hours after delivery. High-dose intravenous vitamin C could, therefore, have a delayed effect that can be detected only in the first days and not hours following delivery. This pilot study, therefore, yields data on both the effectiveness and delay in action of vitamin C in relation to oxidative stress in severe preeclampsia postpartum patients.

Dityrosine, 8-OHdg, and 8-isoprostane have already been studied in pregnancies complicated by preeclampsia or fetal growth restriction. Leon-Reyes et al. reported increased plasma concentrations of dityrosine in women with preeclampsia compared to healthy pregnant controls [28]. Similarly, Scholl et al. found higher urinary concentrations of 8-OHdg in pregnancies complicated by fetal growth restriction, a pregnancy disorder often sharing a common pathophysiology with preeclampsia [29]. Free plasma 8-isoprostane concentrations have also been found to be increased in preeclamptic patients [30,31]. These findings demonstrate an increase in oxidative stress that can induce damage to the vascular endothelium and create maternal complications characteristic of preeclampsia [30,31]. Such complications include pulmonary edema, severe levels of hypertension, eclamptic seizures, and renal dysfunction, which can all occur days or even weeks postpartum [32–34]. Our study was the first to focus on postpartum changes in oxidative stress biomarkers in severe preeclampsia. In addition, we studied the effects of high-dose vitamin C treatment on oxidative stress in this patient population. Our results indicate that increased oxidative stress associated with preeclampsia persists for several days following delivery. Only urinary concentrations of HEL decreased significantly between days one and three after delivery in patients receiving vitamin C. Lower dityrosine and 8-OHdg levels in the vitamin C group at day three after delivery also suggest that high-dose vitamin C reduces oxidative stress in preeclamptic patients after delivery. Prophylactic oral supplementation with antioxidants, including vitamin C, has not been shown to prevent the occurrence of preeclampsia in previous studies [17,18]. Nevertheless, our results suggest the potential benefit of high-dose intravenous vitamin C treatment as a tertiary prevention strategy to reduce excessive oxidative stress, endothelial damage and risks of preeclampsia-related postpartum adverse outcomes. If vitamin C reduces endothelial damage and, consequently, vascular permeability in preeclampsia patients to the same degree that it seems to reduce vascular permeability in other conditions, such as burns, it could reduce the risk of pulmonary complications associated with this disease. At the same time, vitamin C therapy could allow safe additional fluid administration in patients with preeclampsia who are fluid responsive and in whom further increases in preload would be beneficial for preventing end-organ damage such as pre-renal acute kidney injury.

The major strength of this study is its randomized control methodology. This allowed us to account for potential confounding factors that could have an influence on peripartum oxidative stress regardless of vitamin C treatment. Examples of such factors include mode of delivery (vaginal vs. cesarean delivery), gestational age, severity of the disease, dietary influences etc. We used well-validated biomarkers of oxidative stress. Urinary dityrosine, 8-OHdg, 8-isoprostane, and HEL have all been shown to be an adequate tool to assess oxidative stress status outside pregnancy and also in pregnant women [24,25]. Several limitations of this study should also be considered. This was a single-center trial with a small number of patients included. Although urine samples were collected within a randomized setting, major protocol violations occurred with a significant number of patients from

whom samples were not collected or analyzed. Larger studies will, therefore, be needed to confirm or refute our results. Importantly, the outcome studied was not a clinical endpoint, i.e., complications of preeclampsia, but a biochemical one, i.e., urinary values of markers of oxidative stress. Therefore, this should be viewed as a proof-of-concept study showing an enhanced decrease in oxidative stress after delivery in preeclampsia patients receiving vitamin C. Further research is needed to determine whether the decrease in oxidative stress biomarkers seen with vitamin C treatment correlates with the amelioration of the clinical course of severe preeclampsia postpartum.

**Author Contributions:** All authors contributed to the study conception and design. Material preparation, data collection and analysis were performed by M.K., J.O., K.G., K.K., T.F., S.S. and M.L. The first draft of the manuscript was written by M.K. and M.L., and all authors commented on previous versions of the manuscript. All authors read and approved the final manuscript.

**Funding:** This study was funded by a tertiary research project grant from the University Medical Centre Ljubljana (grant No 20170031/ 339 99 06 93) and by the Slovenian Research Agency (research program P3-0124).

**Conflicts of Interest:** The authors declare no conflict of interest. The funders had no role in the design of the study; in the collection, analyses, or interpretation of data; in the writing of the manuscript, or in the decision to publish the results.

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
