# Peer review of "Effect of High-Dose Intravenous Vitamin C on Postpartum Oxidative Stress in Severe Preeclampsia"

_2673-3897, doi:10.3390/reprodmed1020009_

Round 1

Reviewer 1 Report

The study assesses the severity of oxidative stress by measuring the products of non-enzymatic reactions between DNA, proteins, lipids with ROS. The authors tested the concentration of substances in the urine such as: dityrosine, 8-OHdg, 8-isoprostane and HEL at the first and third day after delivery in group treated with vit c and placebo. Patients received vitamin C or placebo in the first 30 minutes after delivery. Urine sample was collected within the first 6 h (could be collected just after vitamin C administration or after 6 h). Therefore, it is not known whether the purpose of the first intake was the baseline assessment of the severity of oxidative stress or the short antioxidative  effect of vitamin C . This solution could have been the cause of the lack of difference between the concentration of the analysed substances on the first and third day. This is the only note for the study. The study was well designed, its strengths are randomization and it blinding: doctors, patients, and researchers. There is no information why 5 patients in the vitamin C group and 7 in the placebo group did not receive urine samples for the study. Was it due to serious condition, lack of diuresis, or just organizational problems? This significantly reduced the size of both groups.

In line 140 is an editorial error. There should be an abbreviation 8-OHdg

Author Response

Dear reviewer,

Sincerely,

Monika Korenč

Reviewer 2 Report

The purpose of the submitted manuscript is to determine whether high-dose intravenous vitamin C reduces oxidative stress postpartum, in patients diagnosed with severe preeclampsia.

The study is well designed, but as the authors identify rightly as a limitation, the sample size is very small. There is little generalizability and yes, it is more appropriate as a proof of concept. It should be described as such from the beginning. If it is proof of concept, more detail should be provided for moving forward in the research, once POC is found successful. Yes, it is interesting, but what can be done with this information?

Minor edits are also necessary and specific examples are line 96, lits should be changed to kits; lines 25, 140, and 142, 8-OGdg should be 8-OHdg; line 118 appears to lack the group descriptor, placebo and; figure 2 should have an l in placebo.

Author Response

Dear reviewer,

Sincerely,

Monika Korenč
